# THE POINT TO WHICH SOFT ACTOR-CRITIC CONVERGES

**Jianfei Ma**
School of Mathematics and Statistics
Northwestern Polytechnical University
`matrixfeeney@gmail.com`

## ABSTRACT

Soft actor-critic is a successful successor over soft Q-learning. While lived under maximum entropy framework, their relationship is still unclear. In this paper, we prove that in the limit they converge to the same solution. This is appealing since it translates the optimization from an arduous to an easier way. The same justification can also be applied to other regularizers such as KL divergence.

## 1 PRELIMINARIES

Consider a regularized infinite-horizon discounted MDP, defined by a tuple $(\mathcal{S}, \mathcal{A}, P, r, \rho_0, \gamma, \Delta)$, where $\mathcal{S}$ is the state space, $\mathcal{A}$ is the action space with finite cardinality $|\mathcal{A}|$, $p : \mathcal{S} \times \mathcal{A} \times \mathcal{S} \to \mathbb{R}$ is the transition probability distribution, $r : \mathcal{S} \times \mathcal{A} \to \mathbb{R}$ is the reward function, assumed to be bounded $\rho_0 : \mathcal{S} \to \mathbb{R}$ is the distribution of the initial state $s_0$, and $\gamma \in [0, 1)$ is the discount factor. We denote $\pi : \mathcal{S} \times \mathcal{A} \to [0, 1]$ as a stochastic policy. We restrict our attention on the regularizer $\Delta : \pi \to \mathbb{R}$, that is, being a function of the policy. Whenever noticed, $\mathcal{H}(s)$ abbreviates the entropy of $\pi(\cdot|s)$.

In SAC Haarnoja et al. (2018), the soft Bellman operator $\mathcal{T}^\pi$ is defined as follows

$$\mathcal{T}^\pi Q(s_t, a_t) = r(s_t, a_t) + \gamma \mathbb{E}_{s_{t+1}}[V(s_{t+1})], \tag{1}$$

where

$$V(s_t) = \mathbb{E}_{a_t \sim \pi}[Q(s_t, a_t) - \eta \log \pi(a_t|s_t)] \tag{2}$$

It is not difficult to see that $\mathcal{T}^\pi$ is a contraction by modifying the reward as $r(s, a) + \gamma \mathbb{E}_{s' \sim p}[\mathcal{H}(s')]$.

With repeatedly applying this operator over an arbitrary starting action-value function $Q$, it approaches to the soft value function.

## 2 CONVERGENCE ANALYSIS

In this section, we firstly study the regularized proxies from an optimization perspective, then state the soft policy iteration, and finally point out the convergence result.

### 2.1 OPTIMIZING WITH REGULARIZATION

Define the regularized state-value function as

$$\tilde{V}^\pi(s) = \mathbb{E}\Big[\sum_{l=0}^{\infty} \gamma^l (r_{t+l} + \eta \Delta_{t+l}) | s_0 = s\Big] \tag{3}$$

where $\eta$ is the temperature parameter, usually positive, determining the relative importance of the regularization term against the reward.

The optimal regularized value function $\tilde{V}^\star(s)$ should satisfy the corresponding optimal Bellman equation[1] for all $s \in \mathcal{S}$

$$\tilde{V}^\star(s) = \sup_\pi \sum_{a \in \mathcal{A}} \pi(a|s) \big[ r(s, a) + \eta \Delta(s) + \gamma \mathbb{E}_{s' \sim p}[\tilde{V}^\star(s')] \big] \tag{4}$$

---

[1]Though we use the summation for simplicity, it can be readily replaced by the integral.

For $\Delta(s) = \mathcal{H}(\pi(\cdot|s))$, we have

**Lemma 1.** *For all $(s, a) \in \mathcal{S} \times \mathcal{A}$, the optimal value function $\tilde{V}^\star(s)$ and the optimal policy $\tilde{\pi}^\star(a|s)$, satisfy*

$$\tilde{V}^\star(s) = \eta \log \sum_{a \in \mathcal{A}} \exp \frac{1}{\eta}\big(r(s,a) + \gamma \mathbb{E}_{s' \sim p}[\tilde{V}^\star(s')]\big)$$

$$\tilde{\pi}^\star(a|s) = \frac{\exp \frac{1}{\eta}\big(r(s,a) + \gamma \mathbb{E}_{s' \sim p}[\tilde{V}^\star(s')]\big)}{\sum_{a \in \mathcal{A}} \exp \frac{1}{\eta}\big(r(s,a) + \gamma \mathbb{E}_{s' \sim p}[\tilde{V}^\star(s')]\big)} \tag{5}$$

from which we define an auxiliary optimal action-value function (not the true one)

$$\tilde{Q}^\star(s,a) = r(s,a) + \gamma \mathbb{E}_{s' \sim p}[\tilde{V}^\star(s')] \tag{6}$$

**Proposition 1.** *For any $V : \mathcal{S} \to \mathbb{R}$ that satisfies $V(s) \leq \tilde{V}^\star(s)$ for all $s \in \mathcal{S}$, then*

$$Q(s,a) \triangleq r(s,a) + \gamma \mathbb{E}_{s' \sim p}[V(s')] \leq \tilde{Q}^\star(s,a) \tag{7}$$

## 2.2 Soft Policy Iteration

Consider the softmax policy class $\Pi$

**Lemma 2.** *(Soft Policy Iteration). Repeatedly application of soft policy evaluation (Haarnoja et al., 2018, Lemma 1) and soft policy improvement (Haarnoja et al., 2018, Lemma 2) to any $\pi \in \Pi$ converges to a policy $\pi^\star$ such that $Q^{\pi^\star}(s,a) \geq Q^\pi(s,a)$ for all $\pi \in \Pi$ and $(s, a) \in \mathcal{S} \times \mathcal{A}$.*

Combining all the aforementioned statements, we formally arrive at

**Theorem 2.** *For any initial policy $\pi_0$ and corresponding action-value function $Q^{\pi_0}$, the convergent points induced by SPI 2 satisfy $Q^{\pi^\star}(s,a) = \tilde{Q}^\star(s,a)$ and $\pi^\star = \tilde{\pi}^\star$.*

*Proof.* The backward direction is obvious as shown in (Haarnoja et al., 2018, proof of Theorem 1), that is, $Q^{\pi^\star} \geq \tilde{Q}^\star$. We only need show the other direction. Since $Q^{\pi^\star}$ is the fixed point of the soft Bellman operator $\mathcal{T}^{\pi^\star}$, thus it must satisfy the Bellman equation as defined in Equation 7 with a value function $V^{\pi^\star}$. And since $V^\star$ is the regularized value function that at most can be obtained, it must have $V^{\pi^\star} \leq V^\star$. By Proposition 1, it follows that $Q^{\pi^\star} \leq \tilde{Q}^\star$. And since $\pi^\star \in \Pi$, it immediately follows that $\pi^\star = \tilde{\pi}^\star$. $\qquad\square$

This theorem connects LogSumExp optimization to policy evaluation and improvement, providing an alternative approach. It links SQL Haarnoja et al. (2017) and SAC, with SAC being superior in optimization. It allows for optimizing regularizers like KL divergence using a different procedure. With a prior $\bar{\pi}$ on the policy, setting $\Delta(s) = -D_{\mathrm{KL}}(\pi|\bar{\pi})$ allows us to derive conservative optimal points and define the conservative Bellman operator, using similar justifications

$$V^{\pi^\star}(s) = \eta \log \sum_{a \in \mathcal{A}} \bar{\pi}(a|s) \exp \frac{1}{\eta}\big(r(s,a) + \gamma \mathbb{E}_{s' \sim p}[V^{\pi^\star}(s')]\big)$$

$$\pi^\star(a|s) = \frac{\bar{\pi}(a|s) \exp \frac{1}{\eta}\big(r(s,a) + \gamma \mathbb{E}_{s' \sim p}[V^{\pi^\star}(s')]\big)}{\sum_{a \in \mathcal{A}} \bar{\pi}(a|s) \exp \frac{1}{\eta}\big(r(s,a) + \gamma \mathbb{E}_{s' \sim p}[V^{\pi^\star}(s')]\big)} \tag{8}$$

$$\mathcal{T}^\pi Q(s_t, a_t) = r(s_t, a_t) + \gamma \mathbb{E}_{s_{t+1}}[V(s_{t+1})],$$

$$V(s_t) = \mathbb{E}_{a_t \sim \pi}[Q(s_t, a_t) - \eta \log \frac{\pi(a_t|s_t)}{\bar{\pi}(a_t|s_t)}] \tag{9}$$

Intervening between policy evaluation based on the conservative Bellman operator, and policy improvement with the softmax policy of the conservative action-value function, we are guaranteed to converge to the optimal policy.

URM STATEMENT

The authors acknowledge that at least one key author of this work meets the URM criteria of ICLR 2023 Tiny Papers Track.

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

## A    PROOF OF LEMMA 1

Following the sketch of Azar et al. (2012), we define the Lagrangian function $\mathcal{L}(s; \lambda) : \mathcal{S} \to \mathbb{R}$

$$\mathcal{L}(s; \lambda) = \sum_{a \in \mathcal{A}} \pi(a|s)\big[r(s,a) + \gamma \mathbb{E}_{s' \sim p}[\tilde{V}^\star(s')]\big] + \eta \mathcal{H}(s) - \lambda(\sum_{a \in \mathcal{A}} \pi(a|s) - 1) \tag{10}$$

Since the objective is linear and $\mathcal{H}$ is strictly-concave in $\pi$, and the probability simplex is at least non-empty, thus slater condition is satisfied, which implies the optimum by solving

$$0 = \frac{\partial \mathcal{L}(s; \lambda)}{\partial \pi(a|s)} = r(s,a) + \gamma \mathbb{E}_{s' \sim p}[\tilde{V}^\star(s')] - \eta \log \pi(a|s) - \eta - \lambda \tag{11}$$

The solution is

$$\pi^\star = \exp\left(-\frac{\lambda}{\eta} - 1\right) \exp \frac{1}{\eta}(r(s,a) + \gamma \mathbb{E}_{s' \sim p}[\tilde{V}^\star(s')]) \tag{12}$$

With the equality constraint

$$\sum_{a \in \mathcal{A}} \pi^\star(a|s) = 1 \tag{13}$$

by applying log transformation on both sides, we can solve for the multiplier as

$$\lambda = \eta \log \sum_{a \in \mathcal{A}} \exp \frac{1}{\eta}\big(r(s,a) + \gamma \mathbb{E}_{s' \sim p}[\tilde{V}^\star(s')]\big) - \eta \tag{14}$$

inserting which into Equation 12, we get

$$\tilde{\pi}^\star(a|s) = \frac{\exp \frac{1}{\eta}\big(r(s,a) + \gamma \mathbb{E}_{s' \sim p}[\tilde{V}^\star(s')]\big)}{\sum_{s \in \mathcal{A}} \exp \frac{1}{\eta}\big(r(s,a) + \gamma \mathbb{E}_{s' \sim p}[\tilde{V}^\star(s')]\big)} \tag{15}$$

And finally plug this result into Equation 4, we get

$$\tilde{V}^\star(s) = \eta \log \sum_{a \in \mathcal{A}} \exp \frac{1}{\eta}\big(r(s,a) + \gamma \mathbb{E}_{s' \sim p}[\tilde{V}^\star(s')]\big) \tag{16}$$

