# OpenReview forum: "The Point to Which Soft Actor-Critic Converges"
_ICLR.cc/2023/TinyPapers — Submitted to Tiny Papers @ ICLR 2023_

### Official Review · Reviewer_gNf7 · 2023-03-21

**Confidence:** 3

**Summary Of Contributions:**

This paper provides theoretical proof that soft actor-critic converges to the same solution as soft Q-learning in the limit. This has the potential to aid and ease optimization and can potentially extend to other regularizers to provide the same benefit.

**Rating:**

High Potential (HP): a submission which meets the reviewing criteria and has potential to make an impact on the field

**Strengths And Weaknesses:**

Strengths:

- Clarity: Relevant literature is discussed, especially those that were used to build the proofs shown in this work.

- Correctness: Proofs appear to be correct. Showing that the Q functions and policies are equal proves equal convergence in the limit.

- Reproducibility: Assumptions and proofs are provided, variables are defined well, and the proofs provided can be followed and reproduced. Preliminaries are well stated.

- Follows basic requirements: Is within the page limits and adheres to the code of conduct

Weaknesses:

- Clarity: While it is demonstrated that these converge to the same solution in the limit, I think it would be great to have some additional details on when the convergence may actually happen and some implications of this result from a real-world training scenario.

**Suggested Changes:**

Thank you for your submission! Overall, I think this paper did a great job making the proposed proofs understandable and clear.

My one suggested change for this paper would be to include more details on the implications of the proofs shown here. I think including some details on how this may impact future training schema and what recommendations you have (e.g., should SAC or SQL be used?) based on what you found here would substantially increase the clarity of the paper.

---

### Official Review · Reviewer_opCg · 2023-03-30

**Confidence:** 4

**Summary Of Contributions:**

This tiny paper uncovers the relation between Soft Q-Learning and Soft Actor-Critic framework optimization based on maximum entropy framework. Convex optimization method is utilized to connect the optimal value function and policy between these two RL frameworks. This analysis method can be further leveraged for other regularizer.

**Rating:**

High Potential (HP): a submission which meets the reviewing criteria and has potential to make an impact on the field

**Strengths And Weaknesses:**

- Strengths:
  - Clarity: The proof process is succinctly and clearly described in the paper, short but efficient enough.
  - Correctness: The analysis strictly follow mathematical analysis with correct theorem in SAC and SQL frameworks, thus the conclusion is correct both theoretically and intuitively.
  - Reproducibility: The lemma and theorem in the paper are all described and proofed in detail. It is easy to reproduce the  conclusion of this paper.
  - Follows basic requirements: This tiny paper follows the all ICLR submit requirements.
- Weaknesses:
  - The writer mentions the  same analysis procedure can also be applied with other regularizers without a deeper illustration .

**Suggested Changes:**

Give an illustration to the generalizability of the proposed theorem.
- Is the process of changing LogSumExp optimization to repeated policy evaluation/improvement generalizable? Can it be applied to transform all the arduous LogSumExp optimization problem into an easier one that is not only in RL setting of SQL/SAC?
- How this introduced analysis procedure can be applied for other specific regularizer like KL divergence. Besides give a specific different form of $\Delta(s)$, how can it help with current RL optimization method?

---

### Comment · Area_Chair_QUGF · 2023-06-06
**Check for Archival**

This work meets the threshold for archival, contents the URM statement and is deanonymized.

---

### Meta-Review · Area_Chair_QUGF · 2023-04-02

**Recommendation:** Invite to present (notable)
**Confidence:** 3

**Metareview:**

This paper theoretically proves soft actor-critic (SAC) and soft Q-learning (SQL) converge to the same solution in the limit. It has the potential to ease optimization and be extended to other regularizers (e.g., KL divergence) to provide the same benefit.

Both reviewers acknowledge the contributions of this paper. Although it is a theory paper, the authors did a great job making the paper concise and understandable. Therefore, the CCR standard is met clearly. The reviewers also give several suggestions for the authors to further improve the paper, such as adding more details on the implications of the theoretical results, and an illustration of the generalizability of the proposed theorem. Please carefully revise and proofread the paper following both reviewers' comments.

Overall, based on the review criteria of the ICLR TinyPaper Track, it meets the CCR standard. The AC believes this paper demonstrates a high quality of research. The theory may inspire further advanced empirical methodology. Therefore, we recommend a notable presentation of this paper. Congrats!

**Summary:**

This paper theoretically proves soft actor-critic (SAC) and soft Q-learning (SQL) converge to the same solution in the limit. It has the potential to ease optimization and be extended to other regularizers (e.g., KL divergence) to provide the same benefit.

**Comments And Feedback To The Authors:**

Please carefully revise and proofread the paper following both reviewers' comments, such as the implications of the theoretical results, and an illustration of the generalizability of the proposed theorem.

**Reason For Not Giving A Higher Recommendation:**

N/A

**Reason For Not Giving A Lower Recommendation:**

* Both reviewers acknowledge the CCR of this paper. It is a clear acceptance case.

* Good paper with the potential to inspire advanced empirical methods.

---

### Decision · Program_Chairs · 2023-04-07

Invite to present (notable)